# Tripartite suppression of fission yeast TORC1 signaling by the GATOR1-Sea3 complex, the TSC complex, and Gcn2 kinase

Tomoyuki Fukuda[1†]*, Fajar Sofyantoro[2,3†], Yen Teng Tai[2], Kim Hou Chia[2‡], Takato Matsuda[2], Takaaki Murase[2], Yuichi Morozumi[2], Hisashi Tatebe[2], Tomotake Kanki[1], Kazuhiro Shiozaki[2,4]*

[1]Department of Cellular Physiology, Niigata University Graduate School of Medical and Dental Sciences, Niigata, Japan; [2]Division of Biological Science, Nara Institute of Science and Technology, Ikoma, Nara, Japan; [3]Faculty of Biology, Universitas Gadjah Mada, Yogyakarta, Indonesia; [4]Department of Microbiology and Molecular Genetics, University of California, Davis, Davis, United States

*For correspondence:
tfukuda@med.niigata-u.ac.jp (TF);
kaz@bs.naist.jp (KS)

[†]These authors contributed equally to this work

Present address: [‡]Cell Cycle Control Group, UCL Cancer Institute, University College London, London, United Kingdom

Competing interests: The authors declare that no competing interests exist.

**Abstract** Mammalian target of rapamycin complex 1 (TORC1) is controlled by the GATOR complex composed of the GATOR1 subcomplex and its inhibitor, the GATOR2 subcomplex, sensitive to amino acid starvation. Previously, we identified fission yeast GATOR1 that prevents deregulated activation of TORC1 (*Chia et al., 2017*). Here, we report identification and characterization of GATOR2 in fission yeast. Unexpectedly, the GATOR2 subunit Sea3, an ortholog of mammalian WDR59, is physically and functionally proximal to GATOR1, rather than GATOR2, attenuating TORC1 activity. The fission yeast GATOR complex is dispensable for TORC1 regulation in response to amino acid starvation, which instead activates the Gcn2 pathway to inhibit TORC1 and induce autophagy. On the other hand, nitrogen starvation suppresses TORC1 through the combined actions of the GATOR1-Sea3 complex, the Gcn2 pathway, and the TSC complex, another conserved TORC1 inhibitor. Thus, multiple, parallel signaling pathways implement negative regulation of TORC1 to ensure proper cellular starvation responses.

## Introduction

Target of rapamycin (TOR) complex 1 (TORC1), a protein kinase complex highly conserved among eukaryotes, is a central regulator of cell growth, proliferation, and metabolism (*González and Hall, 2017*; *Wolfson and Sabatini, 2017*). TORC1 potentiates cellular anabolic processes including protein synthesis by phosphorylating ribosomal protein S6 kinase (S6K) and the eukaryotic translation initiation factor 4E-binding protein (4E-BP) (*Burnett et al., 1998*; *Ma and Blenis, 2009*). On the other hand, TORC1 inhibits the catabolic process autophagy by phosphorylating the components of the autophagy initiation complex (*Kamada et al., 2010*; *Kim et al., 2011*; *Puente et al., 2016*). TORC1 has been extensively studied partly because of its potential as a target for anti-cancer therapies (*Mossmann et al., 2018*). Negative regulators of mammalian TORC1 (mTORC1) are frequently mutated in cancer cells and inhibition of mTORC1 effectively suppresses growth of some human cancers (*Grabiner et al., 2014*; *Saxton and Sabatini, 2017*; *Shimobayashi and Hall, 2016*). Therefore, comprehensive understandings of TORC1 and its regulation are expected to provide critical insights into the molecular pathology of cancers as well as the growth control of normal cells.

TORC1 activity is responsive to diverse physiological and environmental cues, such as growth factors, various forms of stress, energy status, and nutrient availability (*González and Hall, 2017*;

*Wolfson and Sabatini, 2017*). In mammalian cells, the activity of mTORC1 is controlled on the lysosomal surface by the two types of small GTPases, Rheb and Rag. In its GTP-loaded state, Rheb activates mTORC1 (*Saucedo et al., 2003*; *Yang et al., 2017*), and this Rheb function is negatively regulated by the GTPase-activating protein (GAP) activity of the tumor-suppressing TSC complex composed of TSC1, TSC2, and TBC1D7 (*Dibble et al., 2012*). In the absence of growth factors or under energy stress, the TSC complex promotes the hydrolysis of GTP bound to Rheb, leading to mTORC1 inactivation (*Garami et al., 2003*; *Inoki et al., 2003*). The Rag GTPases form a heterodimer of RagA or RagB bound to RagC or RagD, which is anchored to the lysosomal membrane through a scaffold protein complex called Ragulator (*Kim et al., 2008*; *Sancak et al., 2008*; *Sancak et al., 2010*). In the presence of amino acids, RagA/B becomes a GTP-loaded, active form and the Rag–Ragulator complex recruits mTORC1 to lysosomes, facilitating mTORC1 activation by Rheb. RagA/B is negatively regulated by its GAP complex GATOR1 (GAP activity toward the Rag GTPases 1), which is composed of DEPDC5, Nprl2, and Nprl3 (*Bar-Peled et al., 2013*). Amino acid stimuli inactivate the GAP activity of GATOR1 through its inhibitor GATOR2, a complex of the five proteins named WDR24, WDR59, MIOS, SEH1L, and SEC13, leading to an increase in GTP-bound RagA/B and thus, mTORC1 activation (*Sancak et al., 2010*; *Bar-Peled et al., 2013*). The GATOR holocomplex composed of GATOR1 and GATOR2 serves as a conduit to transmit amino acid signals to mTORC1. Leucine and arginine are monitored by Sestrins and CASTOR1, respectively, both of which bind and inhibit GATOR2 to inactivate mTORC1 when cells are starved of these amino acids (*Chantranupong et al., 2016*; *Chantranupong et al., 2014*; *Parmigiani et al., 2014*; *Saxton et al., 2016*). SAMTOR is a sensor for the methionine metabolite *S*-adenosylmethionine and mediates methionine-induced mTORC1 activation through GATOR1 (*Gu et al., 2017*).

The mTORC1 subunits as well as the mTORC1 regulators described above, such as the Rag GTPases, GATOR1, and GATOR2, are conserved from yeast to humans (*Tatebe and Shiozaki, 2017*; *Wolfson and Sabatini, 2017*). The Gtr1–Gtr2 heterodimer, the SEACIT complex, and the SEACAT complex are the budding yeast counterparts of human RagA/B–RagC/D, GATOR1, and GATOR2, respectively (*Dokudovskaya et al., 2011*; *Panchaud et al., 2013*). Based on the structural similarities, the Ego1–Ego2–Ego3 complex is a Ragulator equivalent in budding yeast, whereas no apparent Rheb and TSC orthologs are present in the TORC1 pathway of this yeast species (*Kira et al., 2016*; *Powis et al., 2015*). In the fission yeast *Schizosaccharomyces pombe*, another model eukaryote phylogenetically distant from budding yeast, TORC1 is activated by the Rheb ortholog Rhb1, which is under the negative regulation by the TSC complex as in mammalian cells (*van Slegtenhorst et al., 2004*). In addition, we previously identified GATOR1 and Ragulator counterparts in *S. pombe* and demonstrated that, even under nutrient-rich conditions, TORC1 activity is attenuated by these protein complexes through the Rag GTPase heterodimer Gtr1–Gtr2 (*Chia et al., 2017*; *Fukuda and Shiozaki, 2018*). The GATOR1 function can be replaced by the GDP-locked mutant form of Gtr1, indicating that *S. pombe* GATOR1 functions as GAP for Gtr1. Interestingly, genetic experiments suggest that the negative regulation of TORC1 by the Gtr1–Gtr2 heterodimer is in parallel with that by the TSC complex (*Chia et al., 2017*); thus, the Rag GTPases in *S. pombe* have an ability to regulate TORC1 independent of the Rheb GTPase.

Here, we report the characterization of the *S. pombe* GATOR2 complex, which has been identified based on its physical interaction with GATOR1. Unexpectedly, Sea3, an evolutionarily conserved GATOR2 component, plays an important role in the GATOR1 function to negatively regulate TORC1. In addition, unlike in mammals, the GATOR complex is dispensable for TORC1 regulation in fission yeast under amino acid starvation. We have demonstrated that Gcn2, a protein kinase activated by uncharged tRNAs during amino acid starvation, is yet another negative regulator of TORC1 signaling. On the other hand, TORC1 suppression in the absence of any nitrogen source requires the combined actions of GATOR1, the Gcn2 pathway, and the TSC complex. It is likely that TORC1 activity is regulated by multiple, parallel pathways responsive to different types of nitrogen sources.

## Results

### Identification of GATOR2 in fission yeast

We previously identified the *S. pombe* GATOR1 complex through immunopurification of Npr2, an ortholog of the mammalian GATOR1 component Nprl2, followed by mass spectrometry analysis of the co-purified proteins (*Chia et al., 2017*). In addition to Iml1 and Npr3 that constitute GATOR1, the analysis detected four additional proteins, Sea3, Sea4, Sec13, and Seh1, with sequence similarities to the mammalian GATOR2 components WDR59, MIOS, Sec13, and SEH1L, respectively. To confirm the conservation of GATOR2 that associates with GATOR1, we attempted to improve the purification procedure of GATOR1 by constructing strains where two GATOR1 components were tagged with different epitope tags for affinity purification. From the lysate of strains expressing Iml1 with the *myc* tag and either Npr2 or Npr3 with the FLAG tag, GATOR1 was purified by successive anti-FLAG and anti-*myc* immunoprecipitation procedures and co-purified proteins were identified by mass spectrometry (*Figure 1A*). The GATOR1-interacting proteins included Sea2, an apparent ortholog of the mammalian GATOR2 subunit WDR24, in addition to the four other GATOR2 subunit orthologs that we previously detected (*Figure 1B*). The physical interaction of the five presumed GATOR2 components with GATOR1 was further corroborated by their co-immunoprecipitation with the GATOR1 component Iml1 (*Figure 1C*). Thus, all of the five subunits of GATOR2 appear to be conserved in fission yeast and they form a complex with GATOR1 to constitute the GATOR holocomplex.

Iml1 is localized to vacuolar membranes of fission yeast (*Chia et al., 2017*), and consistently, the other GATOR1 subunits were also detected on vacuoles (*Figure 1—figure supplement 1*). To visualize the cellular localization of GATOR2, its individual subunits were expressed with a C-terminal GFP or GFP-mNeonGreen (GFP-mNG) tag by inserting the GFP- or GFP-mNG-coding sequence to their chromosomal loci. Fluorescence microscopy detected Sea2-GFP-mNG, Sea3-GFP-mNG, and Sea4-GFP on vacuolar membranes that were illuminated by the FM4-64 dye (*Figure 1D,E and F*). Consistent with its role as a component of the nuclear pore complex (*Bai˙ et al., 2004*), Seh1 was observed predominantly around the nuclear periphery, though weaker signals were also detectable on vacuoles (*Figure 1G*). These results are consistent with the notion that, together with GATOR1 and TORC1 (*Chia et al., 2017*), GATOR2 resides on vacuoles. As shown in *Figure 1H*, nuclear periphery signals and punctate fluorescence on the cell cortex were observed with GFP-tagged Sec13, which is known to function as a component of the nuclear pore complex as well as the vesicle coatomer COPII (*Bilokapic and Schwartz, 2012*; *Pryer et al., 1993*; *Siniossoglou et al., 1996*). Therefore, only a small fraction of Sec13 might be in the GATOR2 complex on vacuoles.

### Loss of Sea3 phenocopies the GATOR1 deficiency

To investigate the physiological function of GATOR2 in fission yeast, we constructed strains in which the gene encoding each GATOR2 component was deleted. Sec13 was not included in the analysis as its knockout leads to cell lethality (*Poloni and Simanis, 2002*). Except Sea3, loss of the GATOR2 components resulted in no apparent growth defect even in the presence of the TORC1 inhibitor rapamycin (*Figure 2A*). Thus, inactivation of GATOR2 in fission yeast does not significantly affect the TORC1 activity required for cell proliferation, in contrast to the mammalian GATOR2 that promotes TORC1 activation and cell growth through inhibition of GATOR1 (*Bar-Peled et al., 2013*). On the other hand, cells lacking Sea3 exhibited significantly compromised growth both in the presence ('*sea3Δ*' in *Figure 2A*) and absence ('*sea3Δ seh1Δ sea2Δ sea4Δ*') of the other GATOR2 subunits. Moreover, this *sea3Δ* defect was suppressed by rapamycin in the growth medium (*Figure 2A*, right), implying deregulated activation of TORC1 in cells lacking Sea3. Thus, among the GATOR2 components, Sea3 is likely to have a distinctive role in the negative regulation of TORC1 activity.

The observed *sea3Δ* phenotype rescued by rapamycin resembles those of the GATOR1 defective mutants (*Chia et al., 2017*). Another similarity between the *sea3Δ* mutant and the GATOR1 mutants such as *iml1Δ* is that their growth defects are complemented by the loss of Any1 (*Figure 2B*), an inhibitor of the plasma membrane localization of amino acid permeases (*Nakase et al., 2013*). Therefore, together with the known GATOR1 components, Sea3 appears to suppress deregulated TORC1 activation that impedes cellular amino acid uptake (*Chia et al., 2017*). Thus, we hypothesized that Sea3 functions as part of GATOR1. In accordance with our hypothesis, the growth defects

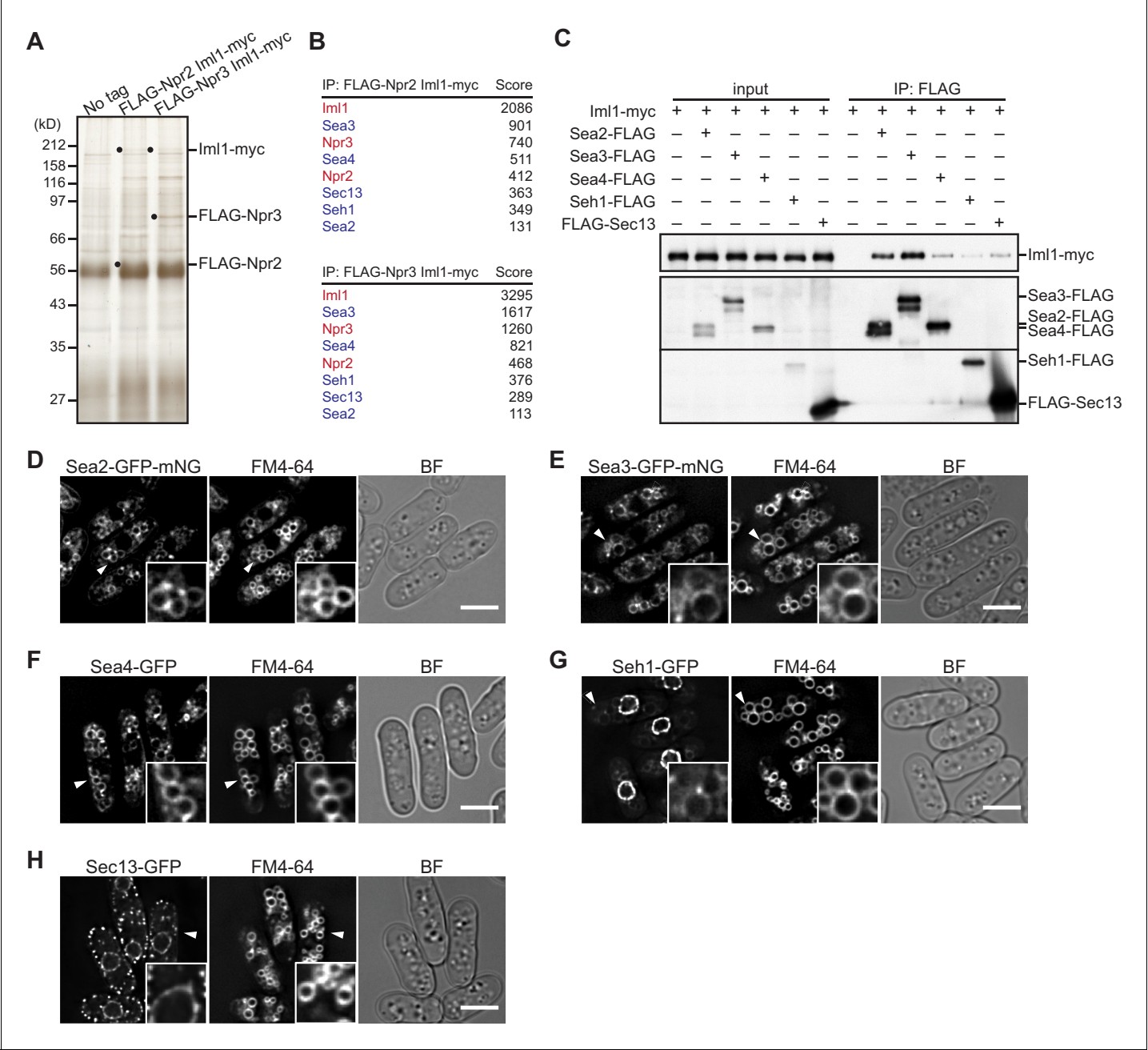

**Figure 1.** Identification of GATOR2 components as GATOR1-binding proteins. (**A**) Affinity purification of the GATOR complex in fission yeast. The GATOR1 complex was purified from cells co-expressing FLAG-Npr2 and Iml1-*myc*, as well as those co-expressing FLAG-Npr3 and Iml1-*myc*, by two successive immunoprecipitation steps using anti-FLAG and anti-*myc* beads. The immunoprecipitates were resolved on SDS–PAGE followed by silver staining. The protein bands corresponding to the tagged Iml1, Npr2, and Npr3 are indicated by black dots. Wild-type cells expressing untagged proteins were used as a control (No tag). (**B**) Proteins co-purified with the GATOR1 subunits in (**A**) were subjected to mass spectrometric analysis. For each protein listed in the tables, two or more peptides were identified. The sum of peptide scores that exceed the 95% confidence level (p<0.05) is also shown for each of the identified proteins. GATOR1 and GATOR2 components are indicated in red and blue, respectively. (**C**) Physical interactions between Iml1 and the GATOR2 components were confirmed. Crude lysates (input) were prepared from the *iml1:myc* strain expressing one of the GATOR2 components tagged with FLAG for immunoprecipitation. The anti-FLAG immunoprecipitates (IP: FLAG) were analyzed by immunoblotting. (**D–H**) The cellular localization of GATOR2 proteins. The indicated fluorescence-tagged strains were grown in Edinburgh minimal medium (EMM) at 30° C for microscopy. Vacuolar membranes were visualized by the fluorescent dye FM4-64. Z-axial images were collected, and mid-section images after deconvolution are shown. Insets show the magnified views of the marked areas. BF, bright-field image. Bars, 5 μm.

The online version of this article includes the following figure supplement(s) for figure 1:

**Figure supplement 1.** The GATOR1 subunits Iml1, Npr2, and Npr3 are localized to vacuolar membranes.

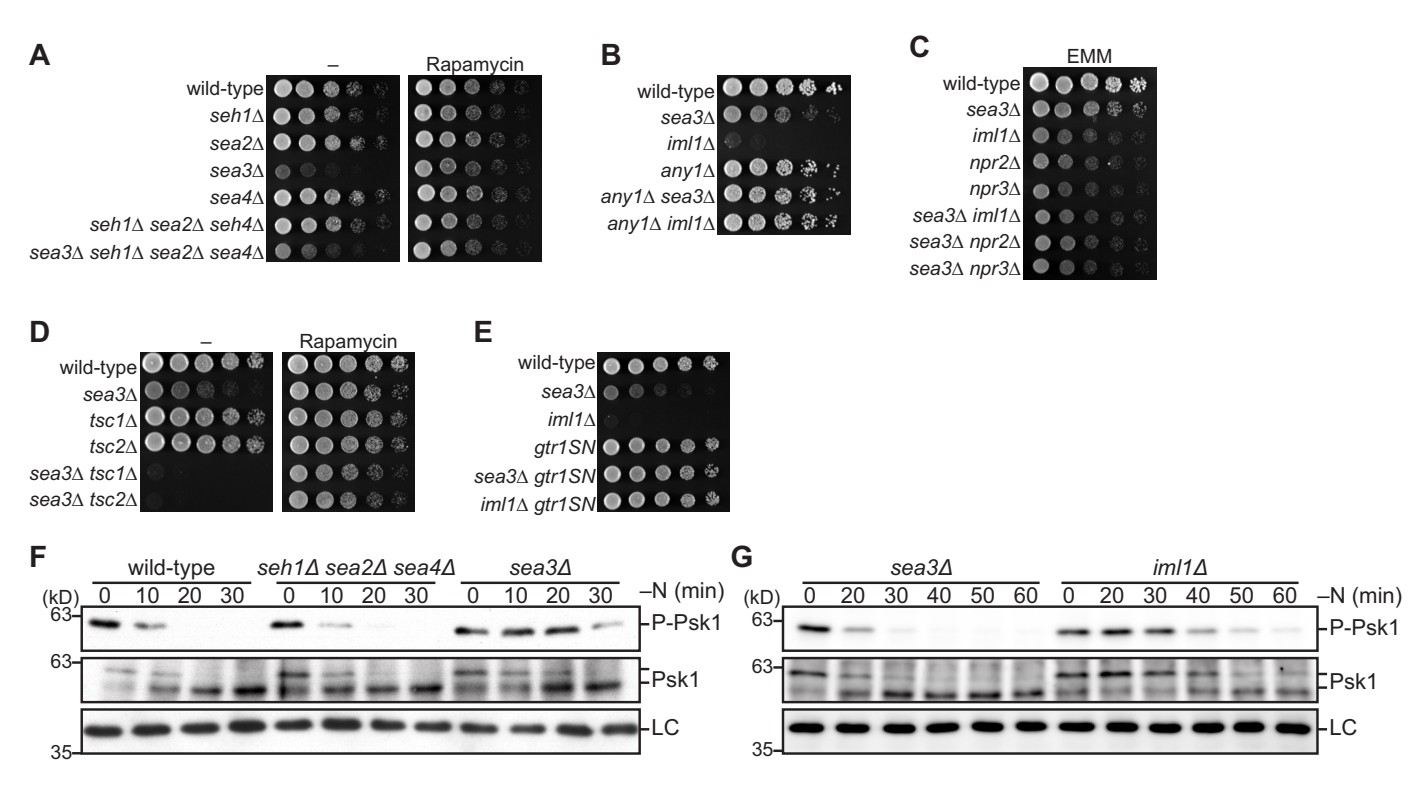

**Figure 2.** Sea3-GATOR1 attenuates TORC1 signaling via the Gtr1–Gtr2 Rag GTPases. (**A–E**) Wild-type and the indicated mutant strains were grown in liquid EMM medium and their serial dilutions were spotted onto the plates. (**A**) Sea3 is required for normal cell growth. The indicated strains were spotted onto solid YES medium with (Rapamycin) or without (–) 100 ng/ml rapamycin. (**B**) Loss of Any1 rescues the growth defect of the *sea3Δ* and *iml1Δ* mutants. The indicated strains were spotted onto solid YES medium. (**C**) Sea3 and the GATOR1 complex function in the same pathway. Growth of the indicated strains was compared on solid EMM medium. (**D**) Sea3 attenuates TORC1 in parallel with the TSC complex. The indicated strains were spotted onto solid YES medium with (Rapamycin) or without (–) 100 ng/ml rapamycin. The growth defect of cells lacking Sea3 was accentuated in the absence of the TSC complex. (**E**) Genetic interactions between Sea3/Iml1 and Gtr1–Gtr2. The indicated strains were spotted onto solid YES medium. The expression of GDP-bound Gtr1 (*gtr1SN*) rescues the growth defect of the *sea3Δ* mutant. (**F, G**) Inactivation of TORC1 upon nitrogen starvation is delayed in cells lacking Sea3. TORC1 activity was monitored by detecting the TORC1-dependent phosphorylation of Psk1 (P-Psk1). The indicated strains were grown in EMM at 30°C, followed by shifting to the same medium without nitrogen source (–N). Cells were collected at the indicated time points after shifting to nitrogen starvation medium for immunoblotting. The samples were also probed with the anti-Psk1 antibody (Psk1), as well as the anti-Spc1 MAPK antibody for a loading control (LC).

of the strains lacking the GATOR1 components (*iml1Δ*, *npr2Δ*, and *npr3Δ*) were not exacerbated by the *sea3Δ* mutation (***Figure 2C***). On the contrary, the *sea3Δ tsc1Δ* and *sea3Δ tsc2Δ* double mutants exhibited severe growth defects suppressible by rapamycin (***Figure 2D***), indicating that, like GATOR1 (***Chia et al., 2017***), Sea3 attenuates TORC1 activity independent of the TSC complex. Furthermore, the loss of Sea3 was complemented by the *gtr1S16N* allele ('*gtr1SN*' in ***Figure 2E***), which expresses the GDP-locked mutant form of the Gtr1 GTPase, suggesting that the function of GATOR1 as Gtr1 GAP is compromised in the absence of Sea3.

To evaluate more directly the role of Sea3 in the regulation of TORC1 activity, we monitored the phosphorylation of the hydrophobic motif in Psk1, a fission yeast S6K1 ortholog (***Nakashima et al., 2012***). In wild-type cells, the phosphorylation of Psk1 disappeared within 20 min after nitrogen starvation, which promptly inactivates TORC1 in fission yeast (***Figure 2F***). The Psk1 phosphorylation in the *sea3Δ* mutant, however, was detectable even after 30 min of nitrogen starvation, indicating deregulated TORC1 activity that lacks an expeditious response to the nutrient. Consistent with the *sea3Δ* phenotype being less severe than that of *iml1Δ* (***Figure 2B,C and E***), dephosphorylation of Psk1 after the starvation in *sea3Δ* cells was not delayed as much as in *iml1Δ* cells (***Figure 2G***); therefore, GATOR1 appears to be partially active even in the absence of Sea3. On the other hand, cells

lacking the other GATOR2 components (*seh1Δ sea2Δ sea4Δ*) displayed starvation-induced Psk1 dephosphorylation comparable to that in wild-type cells (*Figure 2F*).

Taken together, these data strongly suggest that Sea3, a fission yeast ortholog of the mammalian GATOR2 subunit WDR59, promotes the GATOR1 function, which attenuates TORC1 signaling as GAP for the Gtr1 GTPase.

## Sea3 is essential for the interaction between GATOR1 and GATOR2

The unexpected involvement of the GATOR2 subunit Sea3 in the GATOR1 function prompted us to dissect the physical interactions of GATOR1 with Sea3 and the other GATOR2 subunits. Immunoprecipitation of the GATOR1 subunit Iml1 from the *sea3⁺* and *sea3Δ* cell lysates revealed that Sea3 is required for the interaction between Iml1 and the other GATOR2 components Seh1 (*Figure 3A*), Sea2 (*Figure 3B*), Sea4 (*Figure 3C*), and Sec13 (*Figure 3D*). On the other hand, Iml1 and Sea3 were co-immunoprecipitated even in the absence of Seh1, Sea2, and Sea4 (*Figure 3E*), implying that Sea3 directly binds to GATOR1 and anchors the other GATOR2 components to GATOR1. The binding of Sea3 to GATOR1 is dependent on the integrity of the GATOR1 complex, and the absence of any one of the GATOR1 subunits disrupted the Sea3–GATOR1 association (*Figure 3F,G and H*). Consistently, in the absence of intact GATOR1, the vacuolar localization of Sea3 (*Figure 1E*) was lost and the protein diffused throughout the cytosol (*Figure 3I*, *Figure 3—figure supplement 1A*), suggesting that Sea3 is localized to vacuolar membranes through its interaction with GATOR1. It should be noted that deregulated TORC1 activation in the absence of functional GATOR1 per se does not cause the Sea3 diffusion in the cytosol (*Figure 3—figure supplement 1B*); vacuolar localization of Sea3 was observed in cells expressing the GTP-locked Gtr1Q61L mutant protein, which induces TORC1 hyper-activation by mimicking the GATOR1 defect (*Chia et al., 2017*). Furthermore, consistent with the essential role of Sea3 in the interaction between GATOR1 and the other GATOR2 subunits, the vacuolar localization of Sea2, Sea4, and Seh1 (*Figure 1D,F and G*) was abrogated in the *sea3Δ* background (*Figure 3—figure supplement 1C–E*). These results strongly suggest that Sea3 plays an essential role in the assembly of the GATOR holocomplex by mediating the association of the GATOR2 components with GATOR1 on the vacuolar surface.

As described above, the loss of Sea3, but not the other GATOR2 subunits, brings about a growth phenotype similar to that of the GATOR1-defective mutants (*Figure 2*). Therefore, we next examined whether the absence of Sea3 affects GATOR1. Immunoblotting experiments detected no significant difference between wild-type and *sea3Δ* cells in the cellular levels of the GATOR1 components Iml1, Npr2, and Npr3 (*Figure 3—figure supplement 2A*). In addition, the pairwise interactions among the GATOR1 subunits were confirmed by immunoprecipitation experiments with the cell lysate from the *sea3Δ* strains (*Figure 3J,K and L*), suggesting that Sea3 is not essential for the assembly of the GATOR1 complex. We also examined the contribution of Sea3 to the vacuolar localization of GATOR1. As previously found (*Chia et al., 2017*), Iml1 localization to the vacuolar surface was observed in wild-type cells, as well as in cells lacking Npr2 or Npr3 (*Figure 3—figure supplement 2B*). On the contrary, we found that vacuolar localization of Npr2 and Npr3 requires the other GATOR1 components (*Figure 3—figure supplement 2C,D*). In the case of the *sea3Δ* mutant, Iml1, Npr2, and Npr3 were all detectable on vacuoles (*Figure 3—figure supplement 2E,F and G*), indicating that Sea3 is dispensable for the vacuolar localization of GATOR1. On the other hand, immunoprecipitation of the GATOR1 subunit Iml1 found that the physical interaction between GATOR1 and the Gtr1 GTPase was reduced in the *sea3Δ* mutant (*Figure 3M*). This result raises the possibility that Sea3 facilitates the GAP function of GATOR1 by promoting or stabilizing the physical interaction between GATOR1 and the Gtr1 GTPase.

## A conserved arginine in Npr2 is essential for the function of GATOR1

It was proposed that the conserved Arg$^{943}$ of the Iml1 subunit of *S. cerevisiae* GATOR1 functions as an 'arginine finger' essential for the GAP activity of GATOR1 toward the Gtr1 GTPase (*Panchaud et al., 2013*). We previously mutated the corresponding Arg$^{854}$ residue in *S. pombe* Iml1 to alanine and reported that the resultant *iml1R854A* mutant strain showed a growth defect similar to the *iml1Δ* mutant (*Chia et al., 2017*). However, during the course of this study, we found an unexpected mutation in this *iml1R854A* strain and therefore, an *iml1R854A* mutant strain was re-constructed. The new, confirmed *iml1R854A* strain showed no apparent growth defect (*Figure 3—*

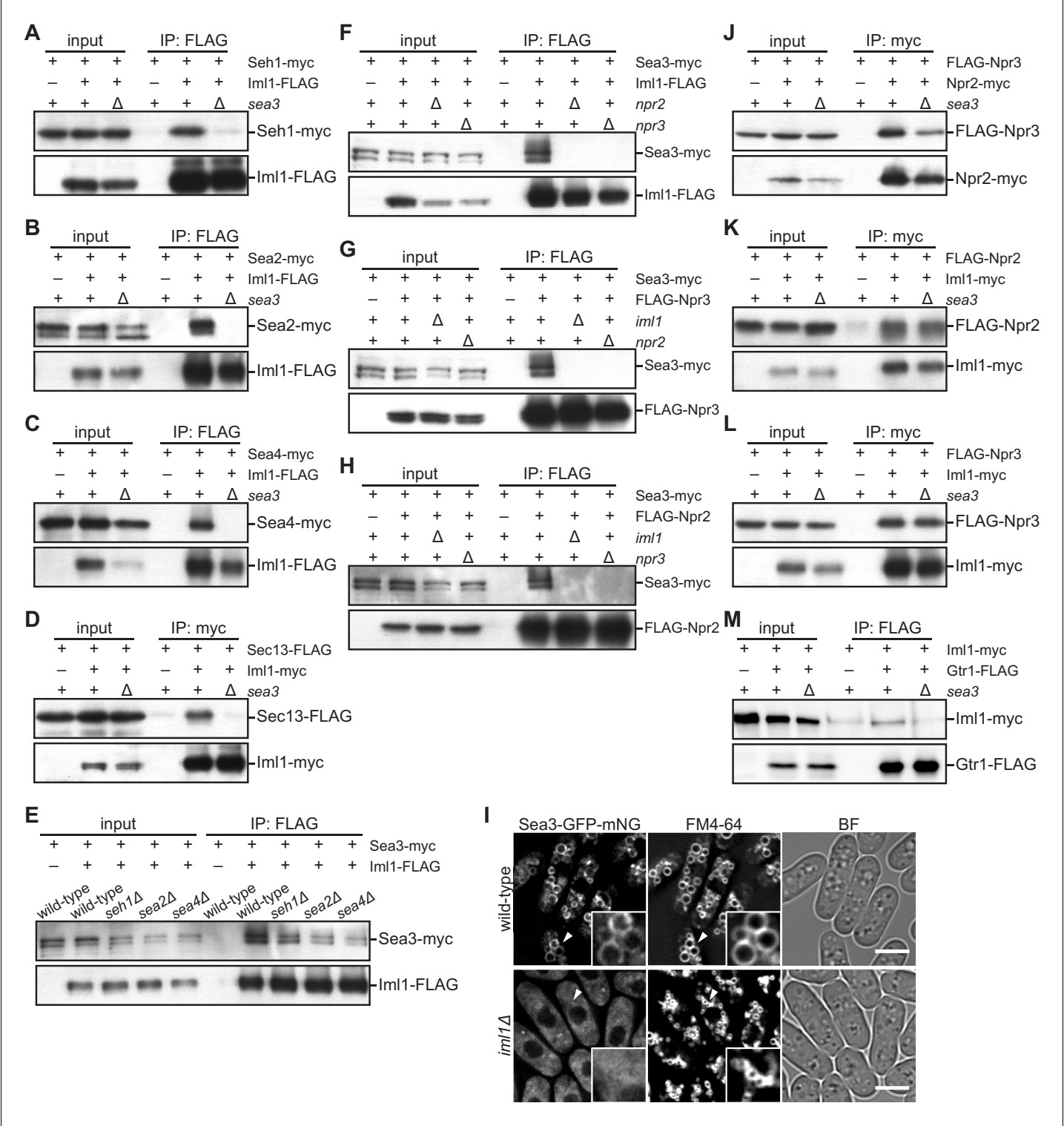

**Figure 3.** Sea3 connects the GATOR2 components to GATOR1. (**A–C**) Sea3 mediates the physical interaction of other GATOR2 proteins with GATOR1. Crude lysates (input) were prepared from cells expressing one of the GATOR2 components tagged with *myc*, with (+) or without (–) co-expression of Iml1-FLAG, in the presence (+) or absence (Δ) of Sea3. The anti-FLAG immunoprecipitates (IP: FLAG) were analyzed by immunoblotting. (**D**) Sea3 mediates the physical interaction of Sec13 with GATOR1. Crude lysates (input) were prepared from *sec13:FLAG* cells, with (+) or without (–) co-expression of Iml1-*myc*, in the presence (+) or absence (Δ) of Sea3. The anti-*myc* immunoprecipitates (IP: *myc*) were analyzed by immunoblotting. (**E**) The interaction of Sea3 with GATOR1 does not require other GATOR2 components. Crude lysates (input) were prepared from *sea3:myc* cells, with (+) or without (–) co-expression of Iml1-FLAG, in wild-type and the GATOR2 mutant backgrounds. The anti-FLAG immunoprecipitates (IP: FLAG) were

*Figure 3 continued on next page*

*Figure 3 continued*

analyzed by immunoblotting. (F–H) The binding of Sea3 to GATOR1 requires an intact GATOR1 complex. Crude lysates (input) were prepared from *sea3:myc* cells, with (+) or without (–) co-expression of Iml1-FLAG (F), FLAG-Npr3 (G), or FLAG-Npr2 (H), in the presence (+) or absence (Δ) of the GATOR1 components (*iml1*, *npr2*, or *npr3*). The anti-FLAG immunoprecipitates (IP: FLAG) were analyzed by immunoblotting. (I) The vacuolar localization of Sea3 is impaired in cells lacking Iml1. The wild-type and *iml1Δ* strains expressing Sea3 tagged with GFP-mNeonGreen were grown in EMM at 30°C for microscopy. The vacuolar membranes were visualized by the fluorescent dye FM4-64. Z-axial images were collected and mid-section images after deconvolution are shown. Insets show the magnified views of the marked areas. BF, bright-field image. Bars, 5 µm. (J–L) Sea3 is dispensable for the integrity of the GATOR1 complex. Crude lysates (input) were prepared from *FLAG:npr3* cells with (+) or without (–) co-expression of Npr2-myc (J) or Iml1-myc (L), as well as from *FLAG:npr2* cells with (+) or without (–) co-expression of Iml1-myc (K). The anti-*myc* immunoprecipitates (IP: myc) were compared between the presence (+) or absence (Δ) of Sea3. (M) Loss of Sea3 reduces the interaction of Iml1 with Gtr1. Crude lysates (input) were prepared from *iml1:myc* cells with (+) or without (–) co-expression of Gtr1-FLAG. The anti-FLAG immunoprecipitates (IP: FLAG) were compared between the presence (+) or absence (Δ) of Sea3.

The online version of this article includes the following figure supplement(s) for figure 3:

**Figure supplement 1.** The GATOR2 components are tethered to the vacuolar surface by GATOR1 complex through Sea3.

**Figure supplement 2.** Sea3 is dispensable for the stability and vacuolar localization of GATOR1.

**Figure supplement 3.** A conserved arginine in Npr2 is required for the GATOR1 function.

---

*figure supplement 3A*), indicating that Arg$^{854}$ of *S. pombe* Iml1 is not essential for the GATOR1 function.

Recently, another conserved arginine residue in mammalian GATOR1, Arg$^{78}$ of the Nprl2 subunit, was proposed to serve as an arginine finger that promotes GTP hydrolysis by RagA/B (*Shen et al., 2019*, *Figure 3—figure supplement 3B*). To assess the role of the equivalent residue in the *S. pombe* GATOR1, Arg$^{98}$ in Npr2 was substituted with alanine to construct an *npr2R98A* mutant strain. The mutant cells exhibited a compromised growth phenotype that was rescued by rapamycin or the *gtr1SN* mutation, an indicative of compromised GAP activity of GATOR1 (*Figure 3—figure supplement 3C*). Though the *npr2R98A* phenotype was not as severe as that of the *npr2* null mutant, these observations are in line with the model that the conserved Arg residue in Npr2, but not the one in Iml1, acts as an arginine finger of GATOR1 also in fission yeast.

## Autophagy induction in the absence of GATOR1 and the TSC complex

The results described above suggest that Sea3, a presumed GATOR2 subunit, contributes to the GATOR1 function that attenuates TORC1 in parallel with Tsc1–Tsc2 (*Figure 2D*, *Chia et al., 2017*). Aiming to evaluate the physiological roles of these negative regulatory mechanisms toward TORC1 activity, we turned our attention to autophagy, which is induced by TORC1 inactivation during nutritional starvation (*Kohda et al., 2007*). Autophagic degradation of GFP-tagged phosphoglycerate kinase (Pgk1) has been successfully used as a quantitative readout of autophagy (*Welter et al., 2010*; *Fukuda et al., 2020*); upon autophagy induction, Pgk1-GFP is transported from the cytoplasm to vacuoles for degradation, but the GFP moiety remains undigested due to its resistance to the vacuolar proteases. Thus, the cellular autophagic activity in response to starvation can be monitored by immunoblotting to detect the free GFP released from Pgk1-GFP (*Figure 4—figure supplement 1A*). As expected, no accumulation of GFP was detected in cells lacking Atg7, a core autophagy regulator, or the vacuolar protease Isp6 (*Kohda et al., 2007*). Using this assay, we examined the nitrogen starvation-induced autophagy in the *tsc2Δ* and *iml1Δ* strains, both of which exhibited only partial reduction in the amounts of released GFP (*Figure 4A*). Though much less than that in wild-type cells, autophagy was still detectable in the *tsc2Δ iml1Δ* double mutant, indicating that, in addition to the GATOR1 and TSC complexes, there must be an additional mechanism to attenuate TORC1 upon nitrogen starvation for autophagy induction.

## The Gcn2 signaling pathway is required for autophagy upon amino acid starvation

The Gcn2 protein kinase is activated in response to amino acid starvation and phosphorylates eIF2α to induce the general amino acid control response (*Zhan et al., 2004*). As Gcn2 is one of the key regulators of the cellular starvation response, we set out to assess its role in autophagy induction. First, in order to test whether amino acid starvation, the inducer of Gcn2 activation, brings about autophagy, leucine-auxotrophic fission yeast cells were shifted to the growth medium without

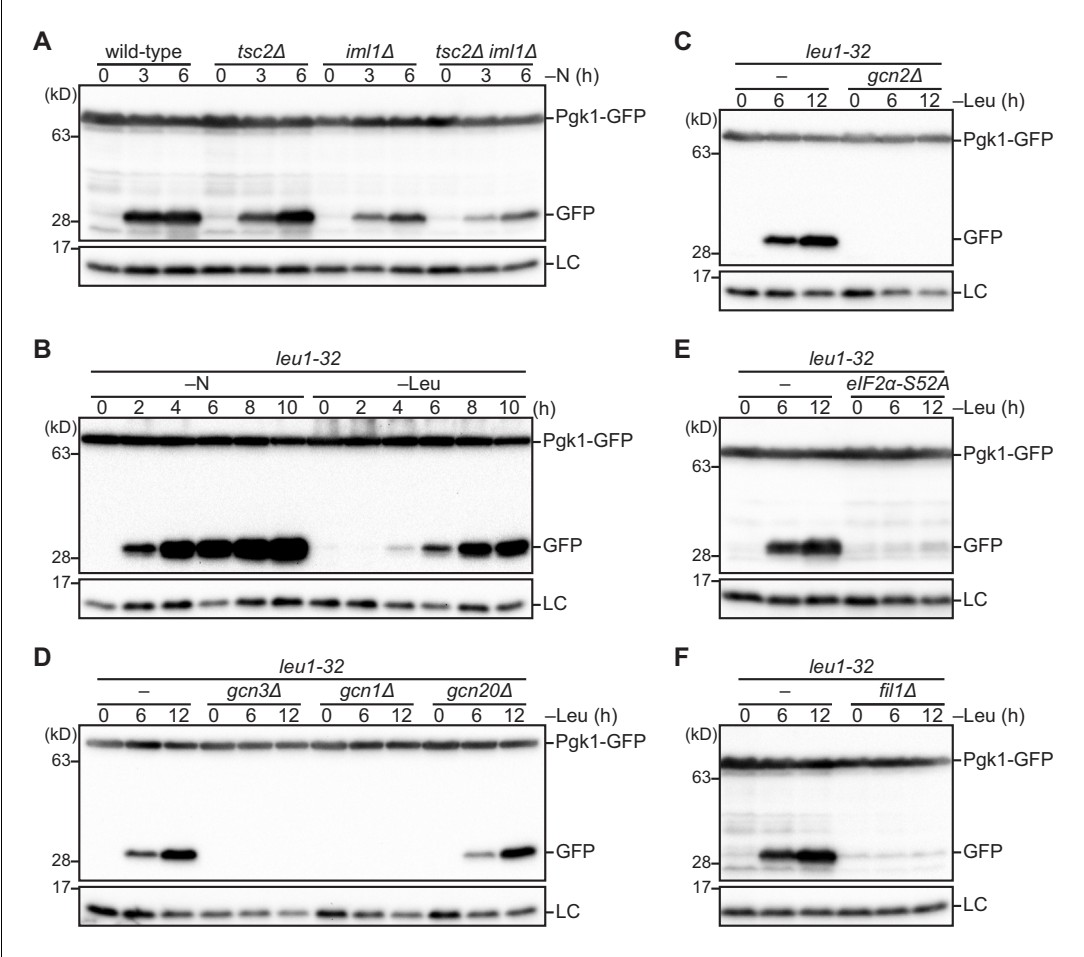

**Figure 4.** The Gcn2 signaling pathway regulates autophagy induction upon amino acid starvation. (**A**) Nitrogen starvation-induced autophagy is reduced but not completely prevented in cells defective in the TSC complex (*tsc2Δ*), GATOR1 (*iml1Δ*), or both (*tsc2Δ iml1Δ*). Autophagy was monitored by detecting the GFP moiety processed from Pgk1-GFP. The indicated strains were grown in EMM at 30°C, followed by shifting to the same medium without nitrogen source (–N). Cells were collected at the indicated time points after shifting to nitrogen starvation medium for immunoblotting against GFP. The samples were also probed with the anti-histone H3 antibody for a loading control (LC). (**B**) Autophagy is induced by leucine starvation or nitrogen starvation. Cells auxotrophic for leucine (*leu1-32*) were grown in EMM supplemented with 1.7 mM leucine at 30°C and shifted to the same medium without nitrogen source (–N) or EMM without the supplement (–Leu). Autophagy was monitored by immunoblotting as in (**A**). (**C–F**) Leucine starvation-induced autophagy requires the Gcn2 signaling pathway. Wild-type and the indicated mutant cells auxotrophic for leucine (*leu1-32*) were collected at the indicated time points after shifting from EMM supplemented with 1.7 mM leucine to EMM without the supplement (–Leu). Autophagy was monitored by immunoblotting as in (**A**). The *eIF2α-S52A* mutant (**E**) carries the Ser$^{52}$ to Ala$^{52}$ mutation in the *tif211$^+$* gene coding eIF2α. The online version of this article includes the following figure supplement(s) for figure 4:

**Figure supplement 1.** The Gcn2 signaling pathway is essential for autophagy induction upon amino acid starvation.

leucine. Such leucine starvation induced Gcn2 activation, which was monitored by immunoblotting to detect phosphorylated eIF2α (*Figure 4—figure supplement 1B*). When cells expressing Pgk1-GFP were starved of leucine, free GFP accumulated, though more slowly than in cells starved of nitrogen (*Figure 4B*); thus, leucine starvation induces autophagy in *S. pombe*. However, no GFP accumulation was detected in *gcn2Δ* cells under leucine starvation (*Figure 4C*), demonstrating that the autophagy induced by leucine starvation is dependent on the Gcn2 kinase. Similarly, in cells of arginine auxotrophy, Gcn2-dependent autophagy was detectable after incubation in the growth medium without arginine (*Figure 4—figure supplement 1C*). Gcn2-dependent induction of autophagy was also observed in cells treated by 3-amino-1,2,4-triazole (3-AT) or methionine sulfoximine (MSX), inhibitors of histidine and glutamine biosynthesis, respectively (*Figure 4—figure supplement 1D,E*). These observations collectively indicate that autophagy can be induced in fission yeast cells

starved of various amino acids in a Gcn2-dependent manner. By contrast, the loss of Hri1 and Hri2, two other eIF2α kinases in *S. pombe* (*Zhan et al., 2004*), did not affect autophagy after leucine starvation (*Figure 4—figure supplement 1F*), indicating that Gcn2 exclusively regulates the leucine starvation-induced autophagy among the *S. pombe* eIF2α kinases.

The Gcn2 kinase forms a complex with the Gcn1 and Gcn20 proteins that facilitate activation of Gcn2 (*Garcia-Barrio et al., 2000*). Autophagy in response to leucine starvation was abrogated by the *gcn1Δ*, but not *gcn20Δ*, mutation (*Figure 4D*), consistent with the fact that Gcn1 directly regulates Gcn2, but Gcn20 has only a minor role in Gcn2 signaling (*Sattlegger et al., 2004*; *Yuan et al., 2017*). The activity of Gcn2 is also known to be positively regulated by Cpc2 in *S. pombe* (*Tarumoto et al., 2013*). Furthermore, in *S. cerevisiae* and mammals, Yih1/IMPACT have been reported as negative regulators of the Gcn2 kinase (*Pereira et al., 2005*; *Sattlegger et al., 2004*). We found that autophagy after leucine starvation was severely impaired in cells lacking Cpc2 and in those overexpressing a fission yeast ortholog of Yih1/IMPACT (*Figure 4—figure supplement 1G*), confirming the essential role of the Gcn2 activity in autophagy induction upon amino acid starvation.

In order to test whether the Gcn2 kinase induces autophagy through phosphorylation of eIF2α, we constructed a strain that expresses eIF2α with its phosphorylation site Ser$^{52}$ substituted by alanine (*eIF2α-S52A*). The strain was defective in the leucine starvation-induced autophagy (*Figure 4E*), indicating the critical role of the eIF2α phosphorylation by Gcn2. Phosphorylated eIF2α inhibits translation in a manner dependent on Gcn3, a component of eukaryotic translation initiation factor 2B (eIF2B) (*Dever et al., 1993*; *Yang and Hinnebusch, 1996*), followed by the up-regulation of the transcription factor Fil1 (*Duncan et al., 2018*). Indeed, we observed that, after leucine starvation, the Fil1 protein increased, which was dependent on Gcn2 and the phosphorylation of eIF2α (*Figure 4—figure supplement 1H*). Moreover, *S. pombe* cells lacking Gcn3 (*Figure 4D*) or Fil1 (*Figure 4F*) displayed autophagy defects during leucine starvation.

Taken together, these results strongly suggest that activation of the transcriptional program induced by the Gcn2-eIF2α-Fil1 pathway promotes autophagy in response to amino acid starvation.

## The Gcn2 signaling pathway regulates TORC1 activity

An obvious question raised by the conclusion above is whether amino acid starvation signaling mediated by the Gcn2-eIF2α-Fil1 pathway induces autophagy through the regulation of TORC1. When fission yeast cells were starved of leucine, the TORC1-dependent phosphorylation of Psk1 became undetectable, indicating that TORC1 activity is suppressed upon leucine starvation (*Figure 5A*). In contrast, Psk1 remained phosphorylated even after the starvation in the *gcn2Δ*, *eIF2α-S52A*, and *fil1Δ* mutant strains (*Figure 5A*), suggesting that TORC1 inactivation in leucine-starved cells is mediated by the Gcn2-eIF2α-Fil1 pathway. Moreover, the autophagy defect of the *gcn2Δ* mutant was complemented by TORC1 inactivation by the TORC1 inhibitors, rapamycin and caffeine (*Figure 5B*). These results suggest that the Gcn2 pathway attenuates TORC1 activity in response to amino acid starvation to induce autophagy.

In stark contrast to the Gcn2 pathway mutants, the strains lacking the functional TSC or GATOR1 complex showed no apparent defect in amino acid starvation-induced autophagy; free GFP, the product of the autophagic degradation of Pgk1-GFP, accumulated after leucine starvation even in the *tsc2Δ iml1Δ* double mutant (*Figure 5C*). Thus, amino acid starvation can induce autophagy independent of the TSC and GATOR complexes, two important negative regulators of TORC1. On the other hand, we observed significant autophagy induction in *gcn2Δ* (*Figure 5D*), *eIF2α-S52A*, and *fil1Δ* (*Figure 5—figure supplement 1*) cells incubated in the growth medium with no nitrogen source. Consistently, nitrogen depletion triggered TORC1 inactivation in these Gcn2 pathway mutants as in the wild-type strain (*Figure 5E*). Therefore, the Gcn2 signaling pathway appears to be specifically responsible for induction of autophagy in response to amino acid starvation. Such a specific role of the Gcn2 pathway during amino acid starvation is also congruous with our observation that the Gcn2-dependent phosphorylation of eIF2α and the Fil1 expression increased more significantly during starvation for leucine than that for nitrogen (*Figure 5F*).

## Gcn2, the TSC complex, and GATOR1 attenuate TORC1 in parallel

As shown above, Gcn2 mediates amino acid starvation signals to inactivate TORC1 and induce autophagy, whereas nitrogen starvation can bring about autophagy even in the absence of Gcn2

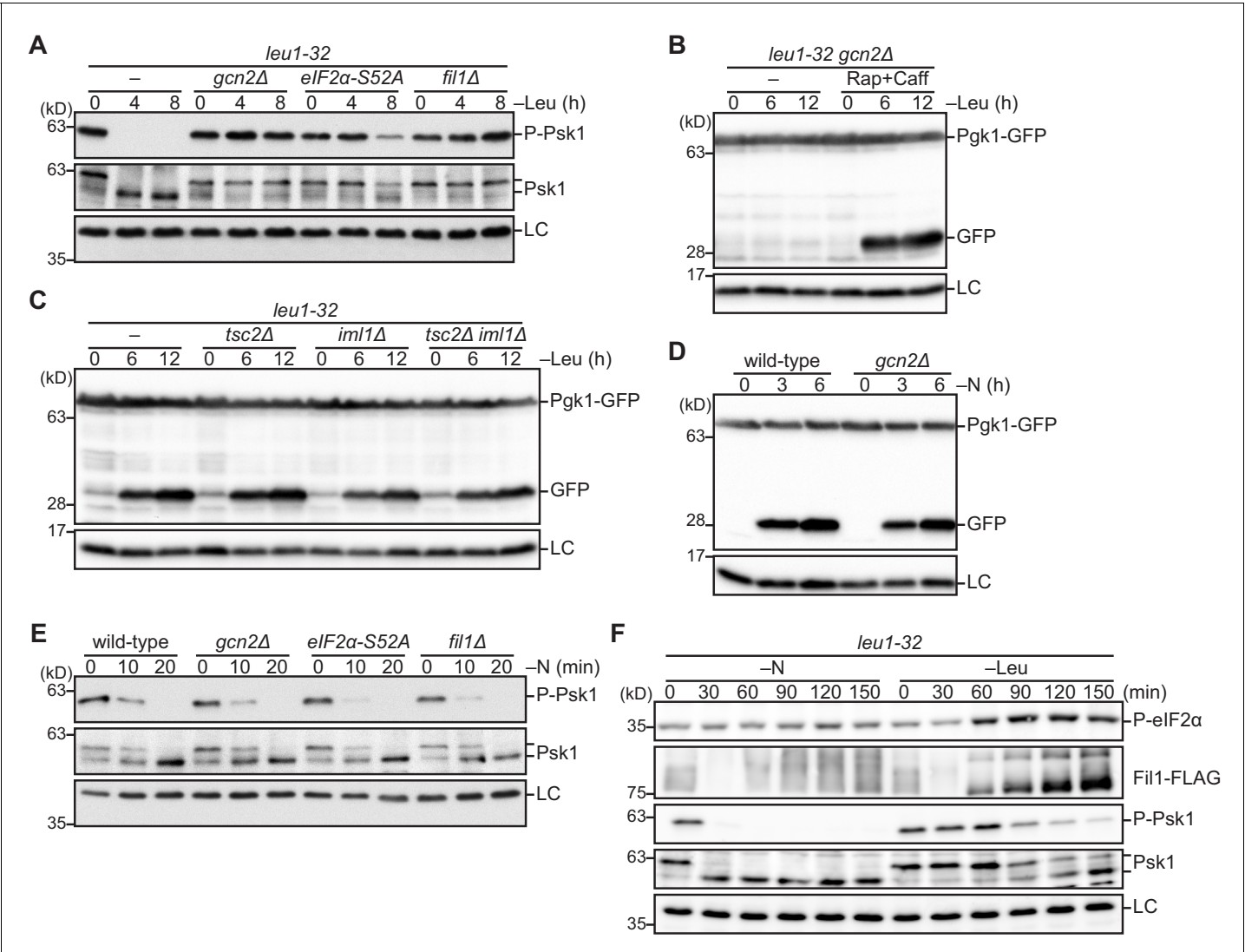

**Figure 5.** The Gcn2 signaling pathway facilitates amino acid starvation-induced autophagy by attenuating TORC1 signaling. (**A**) Inactivation of TORC1 upon leucine starvation is impaired in cells lacking Gcn2 or Fil1 and the mutant carrying the non-phosphorylated form of eIF2α. TORC1 activity was monitored by detecting the TORC1-dependent phosphorylation of Psk1 (P-Psk1). The indicated strains auxotrophic for leucine (*leu1-32*) were grown in EMM supplemented with 1.7 mM leucine at 30°C and shifted to EMM without the supplement (–Leu). Cells were collected at the indicated time points after shifting to leucine starvation medium for immunoblotting. The samples were also probed with the anti-Psk1 antibody (Psk1), as well as the anti-Spc1 MAPK antibody for a loading control (LC). (**B**) Inactivation of TORC1 rescues the autophagy defect of the *gcn2Δ* mutant. Autophagy was monitored by detecting the GFP moiety processed from Pgk1-GFP. The *gcn2Δ* mutant auxotrophic for leucine (*leu1–32*) were grown in EMM supplemented with 1.7 mM leucine at 30°C and shifted to EMM without the supplement (–Leu) with (Rap+Caff) or without (–) addition of 200 nM rapamycin and 10 mM caffeine. Cells were collected at the indicated time points after shifting to leucine starvation medium and subjected to immunoblotting. The samples were also probed with the anti-histone H3 antibody for a loading control (LC). (**C**) The TSC complex and GATOR1 are dispensable for the induction of autophagy upon amino acid starvation. The indicated strains auxotrophic for leucine (*leu1–32*) were subjected to leucine starvation and immunoblotting as in (**B**). (**D**) Gcn2 is not essential for the induction of autophagy upon nitrogen starvation. Wild-type and *gcn2Δ* cells were grown in EMM at 30°C, followed by shifting to the same medium without nitrogen source (–N). Cells were collected and subjected to immunoblotting as in (**B**). (**E**) Inactivation of TORC1 upon nitrogen starvation is independent of the Gcn2 signaling pathway. The indicated strains grown in EMM at 30°C were shifted to the same medium without nitrogen source (–N). Immunoblotting was carried out as in (**A**). (**F**) Phosphorylation of eIF2α and up-regulation of Fil1 are induced by leucine starvation to a higher extent than nitrogen starvation. Cells auxotrophic for leucine (*leu1–32*) were grown in EMM supplemented with 1.7 mM leucine at 30°C and shifted to the same medium without nitrogen source (–N) or EMM without the supplement (–Leu). Immunoblotting was carried out as in (**A**). The samples were also probed with the anti-phospho-eIF2α and anti-FLAG antibodies. The online version of this article includes the following figure supplement(s) for figure 5:

**Figure supplement 1.** The Gcn2 signaling pathway is not necessary for autophagy induction upon nitrogen starvation.

(*Figure 5D*). Therefore, we next examined whether the Gcn2-independent autophagy upon nitrogen starvation is controlled by the other negative regulators of TORC1, such as the TSC and GATOR1 complexes. In the *gcn2Δ tsc2Δ* double-mutant cells starved of nitrogen, autophagy was severely reduced and delayed, in comparison to those in the wild-type and individual single-mutant cells (*Figure 6A*). The autophagy defect in the *gcn2Δ iml1Δ* double mutant was even more severe than that in the *gcn2Δ tsc2Δ* double mutant (*Figure 6B,C*), implying that GATOR1 and Gcn2 are two main contributors to the nitrogen starvation-induced autophagy. In the *gnc2Δ iml1Δ tsc2Δ* triple mutant, a trace of released GFP was detected only after 14 hr of nitrogen starvation (*Figure 6C*), while autophagy takes place within 2 hr in wild-type cells after the starvation (*Figure 4B*). These data indicate that autophagy induction upon nitrogen starvation is dependent on the three negative regulators of TORC1 signaling, GATOR1, Gcn2, and the TSC complex. Not surprisingly, in strains lacking these regulators, severely impaired suppression of TORC1 activity during nitrogen starvation was confirmed by monitoring the TORC1-dependent phosphorylation of Psk1 (*Figure 6D*). Especially, the *gcn2Δ iml1Δ* double mutant exhibited a significant delay in the nitrogen starvation-responsive TORC1 inactivation, consistent with its severe defect in autophagy induction (*Figure 6B*). With a more severe autophagy phenotype (*Figure 6C*), the *gcn2Δ iml1Δ tsc2Δ* triple mutant showed robust Psk1 phosphorylation even after 12 hr of nitrogen starvation (*Figure 6E*). It should be noted that such a significant contribution of Gcn2 to the nitrogen starvation-induced TORC1 inactivation and autophagy was observable only in the absence of the GATOR1 and TSC complexes, but not in their presence (*Figure 5D,E*). Indeed, nitrogen starvation provoked significant eIF2α phosphorylation in the GATOR1- and TSC-defective strains, but not in the wild type (*Figure 6—figure supplement 1A,B*); the loss of the GATOR1 and TSC complexes may be compensated by activation of the Gcn2 kinase to attenuate TORC1. Nitrogen starvation also induced significant Gcn2 activation in autophagy-defective mutants, such as *atg7Δ* and *isp6Δ*, more than in wild-type cells (*Figure 6—figure supplement 1C*), implying that autophagy suppresses Gcn2 activation during nitrogen starvation. It is plausible that reduced autophagy in the absence of the GATOR1 and TSC complexes after nitrogen starvation (*Figure 4A*) may lead to intracellular amino acid depletion, which induces Gcn2 activation via uncharged tRNAs (*Wek et al., 1995*).

To further evaluate the contributions and the kinetics of the individual regulatory mechanisms during nitrogen starvation, we analyzed the *gcn2Δ tsc2Δ*, *iml1Δ tsc2Δ*, and *gcn2Δ iml1Δ* double mutants, each of which has GATOR1, Gcn2, and the TSC complex, respectively, as the sole TORC1 inhibitor. Autophagy became detectable 2–3 hr after nitrogen starvation in the *iml1Δ tsc2Δ* strain, and much later in the *gcn2Δ tsc2Δ* mutant, while *gcn2Δ iml1Δ* cells showed only minuscule autophagy signals even after 9 hr (*Figure 6—figure supplement 2A and B*). Consistently, the reduction in TORC1 activity, as judged by Psk1 dephosphorylation, was also observed early after the starvation in the *iml1Δ tsc2Δ* mutant, followed by the *gcn2Δ tsc2Δ* mutant, and then the *gcn2Δ iml1Δ* strain (*Figure 6D*, *Figure 6—figure supplement 2C,D*). These observations may reflect the contributions of the three negative regulators of TORC1 during nitrogen starvation; Gcn2 signaling contributes the most, whereas the TSC complex does the least among the three. Of note, the *gcn2Δ tsc2Δ* mutant with functional GATOR1 (*Figure 6—figure supplement 2E,F*), but not the *gcn2Δ iml1Δ tsc2Δ* triple mutant (*Figure 6—figure supplement 2F*), exhibited a transient reduction in TORC1 activity within 1 hr of nitrogen starvation. Therefore, it seems that GATOR1 can attenuate TORC1 promptly after the starvation, but negative feedback regulation that re-activates TORC1 or inhibits GATOR1 might be induced in response to the TORC1 attenuation. This GATOR1-dependent TORC1 attenuation preceded the Gcn2-mediated TORC1 inhibition, which became apparent after 2 hr of the starvation (*Figure 6—figure supplement 2E*). In addition, in the *gcn2Δ* single mutant that retains both GATOR1 and the TSC complex, a notable reduction in TORC1 activity was observed even earlier, within 10 min after the starvation (*Figure 6—figure supplement 2G*). Thus, together with GATOR1, the TSC complex may be able to initiate the suppression of TORC1 activity much earlier than the Gcn2 pathway after nitrogen starvation. The relatively late contribution of Gcn2 to TORC1 attenuation in starved cells can be explained by the fact that Gcn2 functions through the Fil1-mediated transcriptional events. Indeed, eIF2α phosphorylation and Fil1 up-regulation were detectable as early as at 15 min after the starvation (*Figure 6—figure supplement 2H*), whereas the Gcn2-dependent TORC1 inactivation took place after 2 hr and later (*Figure 6—figure supplement 2E*).

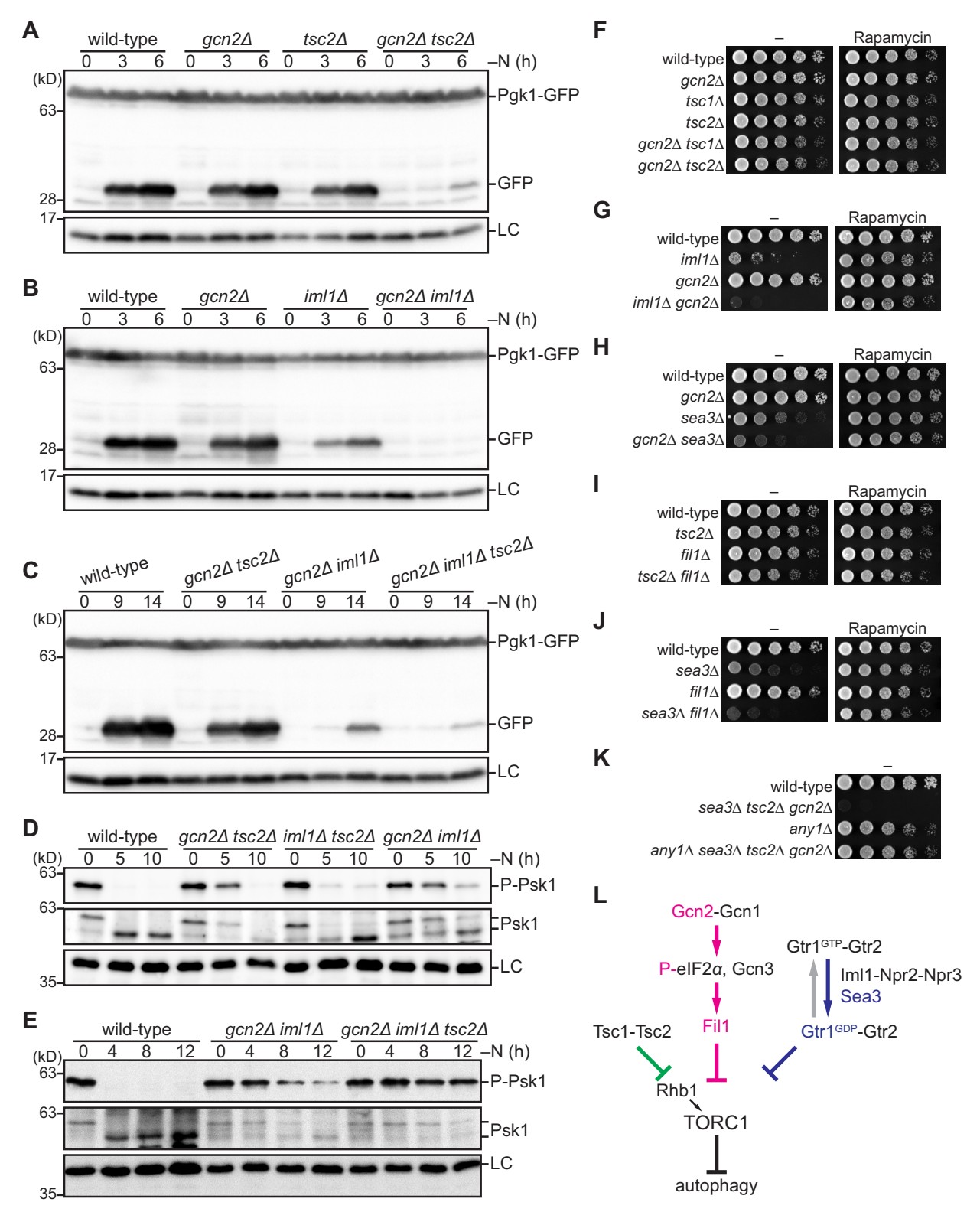

**Figure 6.** The Gcn2 signaling pathway negatively regulates TORC1 upon nitrogen starvation in the absence of the TSC complex and GATOR1. (A–C) Gcn2, Tsc2, and Iml1 promote autophagy induction in parallel upon nitrogen starvation. Autophagy was monitored by detecting the GFP moiety processed from Pgk1-GFP. The indicated strains were grown in EMM at 30℃, followed by nitrogen starvation (–N). Cells were collected at the indicated time points after shifting to nitrogen starvation medium for immunoblotting against GFP. The samples were also probed with the anti-histone H3

*Figure 6 continued*

antibody for a loading control (LC). (D, E) Gcn2, Tsc2, and Iml1 attenuate TORC1 in parallel upon nitrogen starvation. TORC1 activity was monitored by detecting the TORC1-dependent phosphorylation of Psk1 (P-Psk1). The indicated strains cultured in EMM at 30°C were shifted to the same medium without nitrogen source (–N) and collected at the indicated time points for immunoblotting. The samples were also probed with the anti-Psk1 antibody (Psk1), as well as the anti-Spc1 MAPK antibody for a loading control (LC). (F–K) Genetic interactions among the Gcn2 signaling pathway, the TSC complex, and GATOR1. The indicated strains were grown in EMM liquid medium and their serial dilutions were spotted onto solid YES medium with (Rapamycin) or without (–) 100 ng/ml rapamycin. (L) Schematic representation of a model for TORC1 regulation in fission yeast by the Gcn2 signaling pathway, the TSC complex, and GATOR1. Sea3 is involved in the GAP function of GATOR1 toward the Gtr1 GTPase. Gcn2 signaling pathway negatively regulates TORC1 in parallel with the TSC complex and GATOR1. TORC1 inactivation upon amino acid starvation is dependent exclusively on the Gcn2 signaling pathway, whereas that upon nitrogen starvation is mediated redundantly by the three pathways.

The online version of this article includes the following source data and figure supplement(s) for figure 6:

**Figure supplement 1.** The absence of the GATOR1 and TSC complexes during nitrogen starvation up-regulates phosphorylation of eIF2α.
**Figure supplement 2.** Gcn2, GATOR1, and the TSC complex redundantly regulate TORC1.
**Figure supplement 3.** TORC1 inhibitors are involved in cell survival and sexual differentiation during nitrogen starvation.
**Figure supplement 3—source data 1.** Source data for *Figure 6—figure supplement 3B*.

## Roles of the TORC1 negative regulators in the nutritional responses of fission yeast cells

Under nutrient-rich conditions, the TSC and GATOR1 complexes negatively regulate TORC1, moderating TORC1 signaling for optimal growth of fission yeast cells (*Chia et al., 2017*). Therefore, we examined whether the Gcn2 pathway also participates in such negative regulation of TORC1. The *tsc1Δ* and *tsc2Δ* mutants show no apparent growth defect, but introduction of the *gcn2Δ* mutation to them resulted in a detectable growth defect (*Figure 6F*). In addition, the *gcn2Δ* mutation exacerbated the growth defects in cells lacking functional GATOR1 (*Figure 6G*) or Sea3 (*Figure 6H*). Not only *gcn2Δ* but also *fil1Δ* exhibited similar synthetic growth defects when combined with the *tsc2Δ* (*Figure 6I*) or *sea3Δ* (*Figure 6J*) mutation. Importantly, these growth phenotypes caused by the Gcn2 pathway mutations combined with the loss of the other negative regulators of TORC1 were largely suppressed by rapamycin in the growth medium (right panels in *Figure 6F–J*) or by the *any1Δ* mutation (*Figure 6K*). Thus, it is likely that, along with the TSC and GATOR1 complexes, the Gcn2 signaling pathway also contributes to TORC1 attenuation in the presence of ample nutrients.

We also noticed that *iml1Δ* and *tsc2Δ* cells starved of nitrogen gradually lost viability, though such a phenotype was not detectable with the *gcn2Δ* mutant (*Figure 6—figure supplement 3A*). We also assessed the roles of GATOR1, the TSC complex, and Gcn2 in sexual differentiation, a major starvation response in fission yeast (*Mata et al., 2002*). Homothallic (*h⁹⁰*) haploid strains lacking one or more of these TORC1 regulators were starved of nitrogen, and their mating and sporulation efficiencies were measured. Significantly reduced mating and sporulation were observed in strains carrying the *iml1Δ* mutation, suggesting an important role of GATOR1 in the induction of sexual differentiation (*Figure 6—figure supplement 3B*). The *iml1Δ* defect was exacerbated in combination with the *gcn2Δ* or *tsc2Δ* mutation. These observations indicate that the negative regulators of TORC1 are required for proper starvation responses, such as cell survival and sexual differentiation, in fission yeast.

In summary, under nutrient-rich conditions and during starvation, fission yeast TORC1 is negatively regulated by the TSC complex, the Gcn2 signaling pathway, and GATOR1, which requires the presumed GATOR2 component Sea3 for its full function (*Figure 6L*).

## Discussion

In mammalian cells, two classes of Ras-related small GTPases, Rheb and the Rag, serve as key regulators of mTORC1 activation (*González and Hall, 2017*; *Wolfson and Sabatini, 2017*). The TSC and GATOR1 complexes harbor GAP activity toward Rheb and RagA/B, respectively, and control these GTPases to negatively regulate mTORC1 activity. Growth factors and nutrient signals induce mTORC1 activation by suppressing the GAP function of these complexes to promote anabolism and restrain autophagy. We previously identified the GATOR1 complex in fission yeast and showed that the GATOR1 and TSC complexes attenuate TORC1 activity not only during nitrogen starvation but also under nutrient-rich conditions (*Chia et al., 2017*; *Fukuda and Shiozaki, 2018*). We also

detected association of GATOR1 with four additional proteins homologous to the mammalian GATOR2 components, Sea3, Sea4, Sec13, and Seh1 (*Chia et al., 2017*). In addition to them, the improved purification procedure for GATOR1 in this study successfully identified Sea2, an ortholog of the mammalian GATOR2 subunit WDR24, demonstrating the conservation of all the GATOR2 subunits in fission yeast. It has been reported that mammalian GATOR2 is inhibitory to GATOR1, promoting mTORC1 activation in the presence of amino acids (*Bar-Peled et al., 2013*). Unexpectedly, our characterization of Sea3, an ortholog of the mammalian GATOR2 subunit WDR59, implicated Sea3 in the GATOR1 function in fission yeast. First, the *sea3Δ* mutation results in a growth defect to which the loss of GATOR1 is epistatic (*Figure 2C*). Second, suppression of TORC1 by rapamycin alleviates the *sea3Δ* growth defect (*Figure 2A*), as has been found with the other GATOR1 component mutants (*Chia et al., 2017*). Third, expression of the GDP-locked Gtr1 GTPase suppresses the *sea3Δ* phenotype (*Figure 2E*), consistent with the idea that Sea3 contributes to the GATOR1 function as Gtr1 GAP. Fourth, like the GATOR1-defective mutants (*Chia et al., 2017*), cells lacking Sea3 fails to promptly inactivate TORC1 upon nitrogen starvation (*Figure 2F*). Thus, together with Iml1, Npr2, and Npr3, Sea3 appears to function as a fourth subunit of GATOR1 that suppresses TORC1 activity through the regulation of the Gtr1 GTPase. In line with such a notion, Sea3 binds to GATOR1 even in the absence of the other GATOR2 components (*Figure 3E*). Seh1, Sea2, Sea4, and Sec13 can interact with GATOR1 only in the presence of Sea3 (*Figure 3A–D*), implying that Sea3 mediates the interaction between GATOR1 and GATOR2. The predicted molecular architecture of the SEA complex, a GATOR equivalent in budding yeast, suggests that Sea3 also serves as an interaction hub among the constituting subunits (*Algret et al., 2014*), though the *sea3Δ* mutation has little impact on TORC1 activity in this organism (*Panchaud et al., 2013*). We found that *S. pombe* Sea3 is not required for the integrity or vacuolar localization of GATOR1, but promotes the GAP function of GATOR1 by enhancing the interaction between GATOR1 and the Gtr1 GTPase. On the other hand, the loss of Seh1, Sea2, and Sea4 yields no apparent phenotype, providing little clue to the functional relationship between GATOR1 and GATOR2, which are physically linked by Sea3.

Gcn2, an eIF2α kinase conserved among eukaryotes, is activated by uncharged tRNAs that accumulate in amino acid-starved cells (*Wek et al., 1995*). Phosphorylation of eIF2α inactivates translation initiation in general, while enhancing that of specific mRNAs encoding certain transcription factors, such as budding yeast Gcn4, mammalian ATF4, and fission yeast Fil1. Subsequently, they induce transcriptional programs to remedy the starvation stress. Our study has revealed that, in fission yeast, Gcn2 senses lack of different amino acids and induces autophagy through inhibition of TORC1. Phosphorylation of eIF2α followed by an increase in Fil1 is also involved in the Gcn2-mediated autophagy induction, indicating that the Gcn2-eIF2α-Fil1 pathway mediates amino acid starvation signals to negatively regulate TORC1. It should be noted that fission yeast Gcn2 was previously proposed to function downstream of TORC1 because inactivation of TORC1 stimulates Gcn2 kinase (*Valbuena et al., 2012*). Hence, a positive feedback loop may operate during starvation, where Gcn2 signaling suppresses TORC1 and the attenuation of TORC1 further enhances Gcn2 activity. Intricate interplay between TORC1 and Gcn2 signaling in response to different types of stress was also suggested (*Rødland et al., 2014*). Also in budding yeast, Gcn2 has been shown to function both upstream (*Yuan et al., 2017*) and downstream (*Cherkasova and Hinnebusch, 2003*; *Kubota et al., 2003*) of TORC1.

The molecular mechanism by which the Gcn2 pathway attenuates TORC1 is currently unknown. In budding yeast, Gcn2 was proposed to directly bind and phosphorylate one of the TORC1 subunits to inhibit TORC1 activity (*Yuan et al., 2017*). However, a similar scenario is unlikely in fission yeast, where the effectors downstream of Gcn2, such as eIF2α, Gcn3, and the Fil1 transcription factor, are involved in the suppression of TORC1. Interestingly, Gcn4, a Fil1 counterpart in budding yeast, is also known to promote autophagy in response to amino acid starvation (*Ecker et al., 2010*). Moreover, in mammalian cells, GCN2 contributes to mTORC1 suppression during amino acid starvation through the GCN2-activated transcription factor ATF4 (*Ye et al., 2015*). Thus, TORC1 inactivation by the Gcn2-mediated transcriptional program during amino acid starvation is likely to be conserved among diverse eukaryotes. ATF4 induces expression of Sestrin2, which regulates mTORC1 activity via the GATOR complex and the Rag GTPases (*Chantranupong et al., 2014*; *Parmigiani et al., 2014*). On the other hand, our genetic data in this study suggest that the Gcn2 pathway in fission yeast regulates TORC1 independent of the GATOR–Rag pathway (*Figure 6L*), implying a different type of TORC1 regulator(s) as a target of the Fil1 transcription factor. Gcn2 signaling in yeasts is

known to control expression of genes involved in biosynthesis of amino acids, nucleotides, and vitamins, as well as stress responses (*Duncan et al., 2018*; *Natarajan et al., 2001*; *Tarumoto et al., 2013*; *Udagawa et al., 2008*). Therefore, it is also possible that Fil1 does not directly regulate expression of TORC1 regulators; modulation of cellular metabolic, redox, or energy status by the Gcn2-induced transcriptional program may indirectly reduce TORC1 activity in a manner independent of the GATOR1 and TSC complexes.

While TORC1 inactivation in response to amino acid starvation is exclusively mediated by Gcn2, nitrogen starvation suppresses TORC1 through the parallel actions of at least three pathways including GATOR1, the TSC complex, and Gcn2. Among these three pathways, the contribution of Gcn2 is cryptic in the presence of the two others; the Gcn2 pathway mutants exhibit no apparent defect in the TORC1 suppression upon nitrogen starvation (*Figure 5E*). The kinetics of Psk1 dephosphorylation (*Figure 5F*) indicates that nitrogen starvation induces more swift TORC1 inactivation than does amino acid starvation; thus, nitrogen starvation may immediately suppress TORC1 without uncharged tRNA accumulation that activates Gcn2 signaling. Indeed, the Gcn2-dependent eIF2α phosphorylation is not very prominent after nitrogen starvation (*Figure 5F*). The acute TORC1 inactivation upon nitrogen starvation is likely to be mediated by GATOR1 and the TSC complex because, in their absence, TORC1 inactivation is significantly delayed (*Chia et al., 2017*). Importantly, in strains lacking the GATOR1 and TSC complexes, nitrogen starvation leads to robust activation of Gcn2 (*Figure 6—figure supplement 1*), possibly via uncharged tRNAs generated by short supply of amino acids in the absence of robust autophagy in those mutants. In addition, the *gcn2Δ* mutation accentuates the defective TORC1 suppression during nitrogen starvation in strains lacking GATOR1 and the TSC complex (*Figure 6D,E*). Therefore, it is very likely that GATOR1 and the TSC complex are primarily responsible for the nitrogen starvation-induced TORC1 inhibition and that their absence can be compensated by Gcn2 activation, which also suppresses TORC1 activity.

In mammalian cells, leucine and arginine bind to the amino acid sensors Sestrins and CASTOR1, respectively, releasing those sensor proteins from their inhibitory interaction with GATOR2, which then impedes the GATOR1-dependent suppression of mTORC1 (*Chantranupong et al., 2016*; *Chantranupong et al., 2014*; *Parmigiani et al., 2014*; *Saxton et al., 2016*). Sestrins and CASTOR1 are not conserved in fission yeast, and the Gcn2 pathway is instead responsible for the TORC1 attenuation upon leucine and arginine starvation. On the other hand, the GATOR1 and TSC complexes in fission yeast carry out the TORC1 suppression upon nitrogen starvation, though very little is known about how these complexes monitor the nitrogen availability. Thus, Gcn2, GATOR1, and the TSC complex are evolutionarily conserved, but their respective roles in TORC1 regulation may be choreographed differently from organism to organism in order to optimize cellular responses depending on the physiological constraints and requirements. For example, unicellular organisms such as yeast are exposed to dynamic fluctuations in nutritional environment. Fission yeast might have evolved multiple, yet distinct mechanisms that sense nitrogen starvation and induce timely responses in order to properly execute the intricate program of its sexual reproduction (*Mata et al., 2002*). We believe that studies in fission yeast offer an invaluable perspective for comprehensive understanding of conserved and divergent mechanisms of TORC1 regulation in eukaryotes.

## Materials and methods

### Fission yeast strains and general techniques

*S. pombe* strains used in this study are listed in *Supplementary file 1*. Growth media and genetic manipulations for *S. pombe* have been described previously (*Chia et al., 2017*). More than two biological replicates were tested for each experiment. For mating and sporulation assays, homothallic haploid cells were spotted on SSA sporulation plates, incubated at 25°C for 48 hr, and analyzed by microscopy (*Kunitomo et al., 1995*). Fluorescence microscopy analysis was performed as described previously (*Chia et al., 2017*). More than 200 cells were analyzed in each experiment.

Immunoprecipitation, immunoblotting, and mass spectrometry analyses were carried out as described previously (*Chia et al., 2017*). For the Pgk1-GFP processing assay, crude lysates were prepared from cells fixed in 10% trichloroacetic acid by breaking with glass beads in C buffer (8 M urea, 5% SDS, 40 mM Tris–HCl [pH 6.8], 0.1 mM EDTA, and 10% 2-mercaptoethanol). Image data were obtained by the Bio-Rad Chemi Doc XRS imaging system. For immunoprecipitation to detect the

Iml1–Gtr1 interaction, cells were disrupted in the buffer containing 20 mM HEPES–KOH (pH 7.5), 150 mM NaCl, 15 mM $MgCl_2$, 10% glycerol, 0.25% Tween-20, 10 mM sodium fluoride, 10 mM p-nitrophenyl phosphate, 10 mM β-glycerophosphate, 0.1 mM sodium orthovanadate, phenylmethylsulfonyl fluoride (PMSF), and protease inhibitors.

## Key resources table

| Reagent type (species) or resource | Designation | Source or reference | Identifiers | Additional information |
|---|---|---|---|---|
| Genetic reagent (Schizosaccharomyces pombe) | Fission yeast strains used in this study | See Supplementary file 1 | | See Supplementary file 1 |
| Antibody | Anti-phospho-p70 S6K (Thr389) (mouse monoclonal) | Cell Signaling Technology | Cat# 9206; AB_2285392 | (1:4000) |
| Antibody | Anti-Psk1 (rabbit polyclonal) | DOI:10.7554/eLife.30880 | N/A | (1:5000) |
| Antibody | Anti-Spc1 (rabbit polyclonal) | DOI:10.1128/MCB.23.15.5132-5142.2003 | N/A | (1:10,000) |
| Antibody | Anti-FLAG (mouse monoclonal) | Sigma–Aldrich | Cat# F3165; RRID: AB_259529 | (1:4000) |
| Antibody | Anti-c-myc (mouse monoclonal) | Covance | Cat# MMS150P; RRID: AB_291322 | (1:4000) |
| Antibody | Anti-c-myc (rabbit polyclonal) | Santa Cruz Biotechnology | Cat# sc-789; RRID: AB_631274 | (1:4000) |
| Antibody | Anti-GFP (mouse monoclonal) | Takara | Cat# 632380; RRID: AB_10013427 | (1:10,000) |
| Antibody | Anti-histone H3 (rabbit polyclonal) | Abcam | Cat# ab1791; RRID: AB_302613 | (1:10,000) |
| Antibody | Anti-phospho-eIF2alpha (Ser51) (rabbit polyclonal) | Cell Signaling Technology | Cat# 9721; RRID: AB_330951 | (1:2000) |
| Antibody | Anti-DDDDK (rabbit polyclonal) | MBL | Cat# PM020; RRID: AB_591224 | (1:10,000) |
| Antibody | Anti-mouse IgG, Peroxidase conjugated (goat polyclonal) | Merck Millipore | Cat# AP124P; RRID: AB_90456 | (1:10,000) |
| Antibody | Anti-rabbit IgG, Peroxidase conjugated (goat polyclonal) | Jackson ImmunoResearch | Cat# 111-035-003; RRID: AB_2313567 | (1:10,000) |
| Antibody | Anti-rabbit IgG, Peroxidase conjugated (goat polyclonal) | Promega | Cat# W4011; RRID: AB_430833 | (1:2000) |
| Antibody | Anti-mouse IgG Peroxidase conjugated (goat polyclonal) | Promega | Cat# W4021; RRID: AB_430834 | (1:2000) |
| Chemical compound, drug | R-5000 Rapamycin | LC Laboratories | Cat# 53123-88-9 | |
| Chemical compound, drug | Caffeine, anhydrous | Nacalai Tesque | Cat# 06712–42 | |
| Chemical compound, drug | Caffeine | Wako | Cat# 033–06791 | |
| Chemical compound, drug | SynaptoRed C2 (FM4-64) | Biotium | Cat# 70021 | |
| Chemical compound, drug | ViVidFluor Neuro Red (FM4-64) | FUJIFILM | Cat# 222–02121 | |
| Chemical compound, drug | L-Methioninesulfoximine | Nacalai Tesque | Cat# 21730–74 | |
| Chemical compound, drug | 3-Amino-1,2,4-triazole | Wako | Cat# 014–10911 | |
| Commercial assay or kit | SuperSignal West Pico Chemiluminescent Substrate | Thermo Scientific | Cat# 34080 | |
| Commercial assay or kit | SilverQuest staining kit | Invitrogen | Cat# LC6070 | |
| Commercial assay or kit | EzWestLumi plus | Atto | Cat# WSE-7120 | |

*Continued on next page*

*Continued*

| Reagent type (species) or resource | Designation | Source or reference | Identifiers | Additional information |
|---|---|---|---|---|
| Commercial assay or kit | Clarity Max Western ECL Substrate | Bio-Rad | Cat# 1705062 | |

## Acknowledgements
We thank the National BioResource Project/Yeast Genetic Resource Center, Japan, for reagents. We are also grateful to R Kurata for technical assistance.

## Additional information

### Funding

| Funder | Grant reference number | Author |
|---|---|---|
| Japan Society for the Promotion of Science | 17K07330 | Tomoyuki Fukuda |
| Japan Society for the Promotion of Science | 20K06552 | Tomoyuki Fukuda |
| Takeda Science Foundation | | Tomoyuki Fukuda |
| Japan Society for the Promotion of Science | 19K06564 | Yuichi Morozumi |
| Takeda Science Foundation | | Kazuhiro Shiozaki |
| Japan Society for the Promotion of Science | 19H03224 | Kazuhiro Shiozaki |
| Japan Society for the Promotion of Science | 26291024 | Kazuhiro Shiozaki |
| Ohsumi Frontier Science Foundation | 3-0008 | Kazuhiro Shiozaki |
| Sato Yo International Scholarship Foundation | Graduate Student Scholarship | Fajar Sofyantoro |
| Ministry of Education, Culture, Sports, Science and Technology | Graduate Student Scholarship | Yen Teng Tai |

The funders had no role in study design, data collection and interpretation, or the decision to submit the work for publication.

### Author contributions
Tomoyuki Fukuda, Kazuhiro Shiozaki, Conceptualization, Supervision, Funding acquisition, Investigation, Writing - original draft; Fajar Sofyantoro, Investigation, Writing - original draft; Yen Teng Tai, Kim Hou Chia, Takato Matsuda, Takaaki Murase, Hisashi Tatebe, Investigation; Yuichi Morozumi, Funding acquisition, Investigation; Tomotake Kanki, Resources

### Author ORCIDs
Tomoyuki Fukuda (iD) https://orcid.org/0000-0003-2069-7127
Fajar Sofyantoro (iD) https://orcid.org/0000-0003-0952-1956
Kim Hou Chia (iD) https://orcid.org/0000-0002-7958-6635
Tomotake Kanki (iD) http://orcid.org/0000-0001-9646-5379
Kazuhiro Shiozaki (iD) https://orcid.org/0000-0002-0395-5457

## Decision letter and Author response
Decision letter https://doi.org/10.7554/eLife.60969.sa1
Author response https://doi.org/10.7554/eLife.60969.sa2

## Additional files

### Supplementary files
- Supplementary file 1. Fission yeast strains used in this study.
- Transparent reporting form

### Data availability
All data generated or analysed during this study are included in the manuscript and supporting files.

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
