## [Decision Letter]

**Acceptance summary:**

Mammalian Target Of Rapamycin Complex 1 (TORC1) is controlled by the GATOR complex composed of the GATOR1 subcomplex and its inhibitor, the GATOR2 subcomplex sensitive to amino-acid starvation. Fukuda demonstrate that, in fission yeast, the GATOR2 subunit Sea3 (an ortholog of mammalian WDR59) acts with GATOR1 to suppress TORC1 activity. However, GATOR2 is not required for TORC1 activation by amino acid starvation. In contrast, Sea3 does contribute to TORC1 activation caused by nitrogen starvation. These observations clarify the disparate roles of GATOR2 in the fission yeast and mammalian TORC1 pathways. Moreover, the analysis identifies roles for parallel and partially redundant signaling mechanisms that regulate TORC1 in fission yeast.

**Decision letter after peer review:**

Thank you for submitting your article "Tripartite suppression of fission yeast TORC1 signaling by the GATOR1-Sea3 complex, the TSC complex and Gcn2 kinase" for consideration by *eLife*. Your article has been reviewed by three peer reviewers, one of whom is member of our Board of Reviewing Editors, and the evaluation has been overseen by Philip Cole as the Senior Editor. The following individuals involved in review of your submission have agreed to reveal their identity: Kuang Shen (Reviewer #2); Brendan D Manning (Reviewer #3).

The reviewers have discussed the reviews with one another and the Reviewing Editor has drafted this decision to help you prepare a revised submission.

Summary:

In their manuscript, Fukuda et al. identify *S. pombe* Sea3 as a component of the GATOR1 complex that inhibits TORC1 in response to nitrogen deprivation. This protein links the GATOR1 and 2 complexes and is required for the GATOR1-dependent localization of GATOR2 to the vacuolar surface. The authors identify three partially redundant mechanisms that inhibit TORC1 in conditions of nitrogen starvation, Gcn2, the TSC complex, and GATOR1, and determine that only Gcn2 is required to inhibit mTORC1 during starvation of the amino acid leucine.

This manuscript demonstrates the importance of three different nutrient-regulated pathways to suppression of mTORC1 following starvation. The study could be strengthened by a closer consideration of the overlapping functions of these pathways, in particular by addressing their different kinetics (e.g., signaling versus transcriptional responses) and speculation as to why these redundant mechanisms of regulation have evolved relative to mammalian systems.

Essential revisions:

1) The manuscript would be strengthened by a more careful analysis of the kinetics of TORC1 inhibition and induction of autophagy following nitrogen withdrawal at earlier time points. In the double-mutant strains where only one of Gcn2, Iml1, and Tsc2 are expressed, how do loss of P-Psk1, cleavage of Pgk1-GFP, and increased abundance of Fil1 change at early time points following nitrogen deprivation. Do the starvation-sensing pathways act at different times following starvation? The results shown in Figure 6 present data from time points where both post-translational and transcriptional effects would be expected to occur.

2) In Figure 2B and D, *sea3∆* appears to grow better than *iml1∆*, which suggests that loss of Sea3 does not impair GATOR1 function to the same extent as loss of core GATOR1 components. Is this observed when comparing P-Psk1 in these strains following nitrogen starvation as in Figure 2E?

3) One explanation for the inhibition of mTORC1 by Gcn2 during nitrogen starvation is that amino acid levels fall, resulting in activation of Gcn2 via its canonical regulation by uncharged tRNAs in agreement with the findings in Figure 6—figure supplement 1C. This could be tested or at least proposed as a likely mechanism for nitrogen sensing via Gcn2.

---

## [Author Response]

Essential revisions:1) The manuscript would be strengthened by a more careful analysis of the kinetics of TORC1 inhibition and induction of autophagy following nitrogen withdrawal at earlier time points. In the double-mutant strains where only one of Gcn2, Iml1, and Tsc2 are expressed, how do loss of P-Psk1, cleavage of Pgk1-GFP, and increased abundance of Fil1 change at early time points following nitrogen deprivation. Do the starvation-sensing pathways act at different times following starvation? The results shown in Figure 6 present data from time points where both post-translational and transcriptional effects would be expected to occur.

As advised by the comment, we conducted a more careful kinetics analysis of nitrogen starvation experiments using the *iml1∆ tsc2∆*, *gcn2∆ tsc2∆*, and *gcn2∆ iml1∆* double mutants, as shown in new Figure 6—figure supplement 2. The results suggest that Iml1 (GATOR1) and Tsc2 (TSC complex) initiate attenuation of TORC1 promptly after nitrogen starvation (new Figure 6—figure supplement 2G), though the attenuation cannot be sustained much longer in the absence of Gcn2 (“*gcn2∆ tsc2∆*” in new Figure 6—figure supplement 2E). On the other hand, the Gcn2-dependent TORC1 inhibition becomes detectable only after 2 hours of nitrogen starvation (“*iml1∆ tsc2∆*” in new Figure 6—figure supplement 2E). However, as shown in new Figure 6—figure supplement 2H, the Gcn2 kinase is activated within 15 min after nitrogen starvation, followed by increased expression of Fil1. It is conceivable that activation of the Gcn2 kinase does not immediately result in TORC1 attenuation, because TORC1 inhibition by the Gcn2 pathway involves the Fil1-mediated transcriptional program.

2) In Figure 2B and D, sea3∆ appears to grow better than iml1∆, which suggests that loss of Sea3 does not impair GATOR1 function to the same extent as loss of core GATOR1 components. Is this observed when comparing P-Psk1 in these strains following nitrogen starvation as in Figure 2E?

We compared the kinetics of Psk1 dephosphorylation between *sea3∆* and *iml1∆* during nitrogen starvation (new Figure 2G). The starvation-induced inactivation of TORC1 in the *sea3∆* mutant was not compromised as much as that in the *iml1∆* mutant, consistent with the notion that loss of Sea3 does not impair the GATOR1 function to the same extent as loss of the core GATOR1 components.

3) One explanation for the inhibition of mTORC1 by Gcn2 during nitrogen starvation is that amino acid levels fall, resulting in activation of Gcn2 via its canonical regulation by uncharged tRNAs in agreement with the findings in Figure 6—figure supplement 1C. This could be tested or at least proposed as a likely mechanism for nitrogen sensing via Gcn2.

We agree with the reviewer’s view that activation of Gcn2 signaling during nitrogen starvation is triggered by uncharged tRNAs, and have included such discussion in the revised manuscript as suggested (e.g. subsection “Gcn2, the TSC complex, and GATOR1 attenuate TORC1 in parallel” and Discussion). We have also included new data showing that Hri1 and Hri2, two other eIF2α kinases mainly responsive to heat shock and arsenic stress (Zhan et al., 2004, cited in the manuscript), are not involved in autophagy induction upon nitrogen starvation (new Figure 4—figure supplement 1F). The results emphasize the specific role of the Gcn2 kinase in the starvation-induced eIF2α phosphorylation that leads to TORC1 attenuation.